# A Review: Evolution and Diversity of Optical Fibre Plasmonic Sensors

**DOI:** 10.3390/s19224874

**Published:** 2019-11-08

**Authors:** Thomas Allsop, Ron Neal

**Affiliations:** 1School of Engineering and Computer Science, University of Hull, Hull HU6 7RX, UK; 2School of Computing, Communications and Electronics, University of Plymouth, Plymouth PL4 8AA, UK; R.Neal@Plymouth.ac.uk

**Keywords:** optical fiber sensors, plasmonic nanostructures, bio-sensors, gratings

## Abstract

The purpose of this review is to bring to the attention of the wider research community how two quite different optical sensory techniques were integrated resulting in a sensor device of exceptional sensitivity with wide ranging capability. Both authors have collaborated over a 20 year period, each researching initially surface plasmon resonance (SPR) and optical fibre Bragg grating devices. Our individual research, funded in part by EPSRC and industry into these two areas, converged, resulting in a device that combined the ultra-sensitive working platform of SPR behavior with that of fibre Bragg grating development, which provided a simple method for SPR excitation. During this period, they developed a new approach to the fabrication of nano-structured metal coatings for plasmonic devices and demonstrated on fibre optic platform, which has created an ultra-sensitive optical sensing platform. Both authors believe that the convergence of these two areas will create opportunities in detection and sensing yet to be realised. Furthermore, giving the reader “sign-post” research articles to help to construct models to design sensors and to understand their experimental results.

## 1. Introduction

Optical fibre sensors have been researched and developed over the last four decades. The earliest dealt with measuring strain and temperature using the physical and material properties of the fibre itself [1,2]. This initial research activity rapidly expanded in the late 1980s as a direct result of the invention of the fibre Bragg grating (FBG) [3,4]. FBGs are comb-like corrugations in the refractive index of the fibre core, formed by photobleaching the Ge-Doped core glass. At a specific wavelength the grating behaves as a resonator, a consequence of a “phase matching condition”. The grating reflects the light at this wavelength and is transparent to other wavelengths [5].

This discovery provided a springboard for a wide range of research activity that led to a a number of fibre sensors using different grating structures, for example, long period gratings, (LPG) and tilted gratings [6,7]. In parallel with this fibre grating-based sensor development, research into surface plasmon resonance, (SPR), was indicating the promise of highly sensitive biological and chemical sensors [8].

Surface plasmon polaritons, (SPP), are defined as electromagnetic waves, coupled with free electron charge oscillations that occur in a metal close to the metal surface [9]. Under the right conditions a resonance is set up resulting in an electromagnetic evanescent wave riding on the metal surface, that decays into the surrounding dielectric medium. Any change in the dielectric medium modifies the properties of the SPP. At optical frequencies, this phenomenon is referred as Surface Plasmon Resonance (SPR) [9,10]. A more detailed description of plasmonic behaviour is covered later in this review. Surface plasmons have been studied and explored over several decades using Knetschmann and Otto configurations [11]. These studies indicated that the surface wave was extremely sensitive to any minute change in the dielectric space occupied by the plasmonic wave. It was then apparent that SPR could be used to monitor processes and to the detection of substances introduced onto the metal surface. The first optical fibre SPR sensor was demonstrated in the early 1980s [8] as a gas sensing device. Since then, SPR sensing applications and schemes have played an important role in applied optics, particularly in bio and chemical sensing [12]. Most of the research effort into SPR devices made use of visible and near infrared radiation, 480 nm to 900 nm [13]. More recently from 1 μm to longer wavelengths [14]. Research has also taken place into the effects of different surface topologies, using mainly gold, on the spectral characteristics of the plasmons. This work has led to numerous publications dealing with short range plasmons, (SSP), long range plasmons (LSSP) and localised plasmons (LSP) [15,16,17]. The ultra-high sensitivity that plasmonic sensors can achieve has triggered research into biosensing applications such as biomolecular interaction with a bio-recognition molecule [18]. This research indicates that a few molecule detection systems are achievable and a stochastic for normal diffusion binding kinetic reactions.

In this review, we will cover the fundamentals and methods of plasmonic sensing and the way in which the various parameters affect the spectral performance. In addition, we will focus on the different strategies that are being used to achieve optimum performance and characterisation. Completing the review by giving an appraise of the current issues of plasmonic sensing applications and schemes. Finally, both authors realise that photonic grating technologies and techniques coupled with the growth in the materials for optical plasmonic behavior is giving rise to new opportunities for researchers in new fields and new applications. It is hoped that this review will suggest to researchers signpost articles for them to understand the underlying physics and assist them to create new fibre optic devices.

## 2. Materials and Methods Principles of Surface Plasmons

Although there are a number of different methods available to enable the excitation of surface plasmons, the following treatment is applicable to whatever method is used. We have chosen the free-space excitation method as it provides a very clear visualisation of the physical processes involved. Figure 1 shows a planar glass waveguide, the high-index prism necessary to couple the incident light into and out of the waveguide and the evanescent plasmon field riding on the surface of the metal film. The resonance condition for this to occur depends on the wavelength of the incident light, the angle of incidence and the polarisation angle of the light. When these conditions are satisfied there is a change in the output from the detector [19]. 

There are several factor that govern the spectral index sensitivity that underlines the performance of all the sensors created from surface plasmons. The plasmons exist at a metal–dielectric interface and obey the following dispersion relation for two homogeneous semi-infinite media [20]. The propagation constant $\beta {\left(\lambda \right)}_{spp}$ of the plasmon field is totally dependent on the permittivity (refractive index) of the media above and in contact with the metal film. The following mathematical modelling of the action is generally applicable to whatever waveguide is used, for example an optical fibre waveguide. Equation (1) gives the dispersion relation for two homogeneous semi-infinite media [10,20]:(1)$$\beta {\left(\lambda \right)}_{spp}=k\sqrt{\frac{\varepsilon_{m}\left(\lambda \right)n_{s}{\left(\lambda \right)}^{2}}{\varepsilon_{m}\left(\lambda \right)+n_{s}{\left(\lambda \right)}^{2}}}$$
where *k* is the free space wave number, *ε_m_* is the dielectric constant of the metal (*ε_m_* = *ε_mr_*+ *i*
*ε_mi_*) and *n_s_* is the refractive index of the dielectric sample to be tested. The change in the detector output at resonance, which is produced in the reflection spectrum as shown in Figure 1 defines the spectral sensitivity and is described by Equation (2) [10,20]:(2)2πλnpsin(θ)=Re[β(λ)spp]

Using Equations (1) and (2), the waveguide propagation modes can be estimated. There are available, several methods for doing this depending on the complexity of the waveguide spatial geometry and the differing refractive indices involved. A general approach is to use conformal mapping techniques [21] that allows simplification of the waveguide spatial geometry and then to apply Yeh’s algorithm method [22,23] to account for the different index layers, with outer most layer representing the surrounding medium. Using this detail, the various waveguide modes can be identified using the dispersion relationship that can be supported by the waveguide/optical fibre. What is required now is to investigate the spectral behaviour of the plasmonic device as a function of refractive index of the surrounding medium, there are many procedures in the literature to achieve this goal. One of the easier ways is to implement the Fresnel’s equations for a layered system (depending on coating structure being modelled) for different refractive indices of the surrounding medium, calculating the reflected intensity of the for p-polarised light. REF 10 gives the quantitative description of the minimum of the reflected intensity R for a SPR:(3)$$R={\left|\frac{E_{r}^{P}}{E_{0}^{P}}\right|}^{2}={\left|\frac{r_{n_{w}n_{m}}^{P}+r_{n_{m}n_{s}}^{P}\cdot \exp\left(2ik_{z}\left(n_{m}\right)d\right)}{1+r_{n_{w}n_{m}}^{P}\cdot r_{n_{m}n_{s}}^{P}\cdot \exp\left(2ik_{z}\left(n_{m}\right)d\right)}\right|}^{2}$$
where $E_{0}^{P}$ is the incident and $E_{r}^{P}$ is the reflected field, *d* is the thickness of the metal film and $r_{i,j}^{P}$ are the P-polarisation amplitude reflection coefficients between layers *i* and *j,* is defined as $r_{i,j}^{P}=\left(\frac{k_{z,i}}{\varepsilon_{i}}-\frac{k_{z,j}}{\varepsilon_{j}}\right)/\left(\frac{k_{z,i}}{\varepsilon_{i}}+\frac{k_{z,j}}{\varepsilon_{j}}\right)$ where $k_{z,j}$ are the wave vector components of the incident light normal to each layer and *ε_j_* are the permittivities of the *j*^th^ layer, *n_m_* is the index of the metal film, *n_s_* is the effective index of the surrounding medium and *n_w_* is the index of the waveguide/optical fibre. The wave vector components $k_{z,j}$ are determined from the calculations of the modes for the metal-coated waveguide or optical fibre. The procedure out-lined above will yield, a transmission curve of the surface plasmon, see Figure 1. 

From the above modelling, an estimation of the refractive index sensitivity of the surface plasmons can be calculated and a Figure of Merit (FOM) [24], usually referred to as the bulk-refractive index sensitivity. This definition depends on four principle parameters. These are the wavelength of the incident illumination, θ the angle of incidence, commonly called the angular sensitivity, the wavelength shift of the SPR, defined as the spectral sensitivity, the optical strength of the excitation intensity and lastly the phase of the p-polarised component defined as the phase sensitivity [25]. In this review, we have selected to use the spectral sensitivity as we consider this to be the most useful performance indicator, which can be accurately measured, providing an FOM that allows interested groups to compare the merits of other plasmonic device schemes and which is most appropriate for our work using optical fibre waveguides. These are defined, with due consideration to Figure 1, as:
(4)$$S_{\lambda B}=\frac{\Delta \lambda }{\Delta n_{s}}\;\mathrm{and}\;{\mathrm{FOM}}_{B}=\frac{S_{\lambda B}}{\Delta \lambda_{sPP}}$$

The other important parameter is the refractive index resolution and the limit of detection. The index resolution of a plasmonic sensor is defined as the smallest measurable change that can be monitored during an experiment over the intrinsic noise of the plasmonic sensing scheme over the detectable range. This quantity is expressed in terms standard deviation of noise of the sensor output σ_SO_ and bulk sensitivity *S**_λB_* [24], see below. This is closely related to the limit of detection, which is the smallest changeable amount over and above the intrinsic noise. It depends on the measurement instrumentation (light source spectral width, which leads to a finite wavelength resolution of light source, and system noise characteristics) as well as on the noise bandwidth in a measurement, see below [26]. *N* stands for the number of discrete spectral data points that comprises a spectral scan across the resonant peak and $\Delta \lambda_{\mathrm{error}}$ directly relates to spectral noise and *σ_SO_*:
(5)$$n_{noise}=\frac{\sigma_{SO}}{S_{\lambda B}}\;\mathrm{and}\;n_{LOD}=\sqrt{\frac{2\cdot \left(\Delta \lambda_{sPP}+\Delta \lambda_{\mathrm{error}}\right)}{N}}/S_{\lambda B}$$

These parameters are universally used to characterise the performance of plasmonic sensors. There are additional issues that need to be taken into account concerning *n_s_*, the refractive index of the sample under investigation. This is dependent upon the spatial extension of the evanescent field into the sample and yields a “sensing volume” [27]. There are two main parameters that dictate the spatial extension of the evanescent field, these are the materials optical constants forming the metal–dielectric interface and the wavelength of the surface plasmon [28], the analytical expression for the spatial extension into the surrounding medium *n_s_* is given below; *Z_spp_*. Furthermore, the interaction length to the surrounding environment is dictated by the propagation length of the surface plasmon and is also given below; *L_sp_* is the decreases in intensity to 1/e [14], this assumes a smooth surface. To optimise a surface plasmon for a specific wavelength then the thickness of the various materials that constitute the coatings/films need to be taken in consideration [29]. The authors are not going to develop the complete design algorithm for a conventional surface plasmon but to bring to the attention of the reader which parameters are important to determine their values and relevant literature references for the reader to produce a model of the spectral behaviour of the conventional plasmonic device:
(6)Zspp=Im[εm(λ)+ns(λ)2·λ4πns(λ)2] and Lsp=Re[εm(λ)]·λ2π(Re[εm(λ)]·ns(λ)2Re[εm(λ)]+ns(λ)2)32·Im[εm(λ)]

This becomes important when you are detecting a substance within a solution, thus giving a bulk-refractive index sensitivity. The aim of the detection scheme is to detect a specific bimolecular reaction, is achieved by using recognition molecule [30]. Therefore, to maximise the surface plasmon sensitivity is to ensure that the majority or all the “sensing volume” is occupied by the recognition molecule and its target molecule, this is illustrated in Figure 2. This is also referred to has the “surface refractive index sensitivity” [27] *S_S_*, this becomes an important physical characteristic of a type of surface plasmon, a localised surface plasmon.

Figure 2a shows the sensing volume; *V_s_*, that will detect changes in a solution, such as salinity. In Figure 2b, the metal film is covered by a recognition molecule, so the only volume of interest is *V_s_* but the evanescent field fills a volume *V_t_* thus the sensitivity of the detector diminishes:
(7)$$S_{S}=\frac{\Delta \lambda }{\Delta n_{Surf}}$$

In this type of sensor called a “plasmonic affinity sensor” the capture of the target molecule occurs within a smaller volume and is in close vicinity to the metal film rather than the full volume of the surface plasmon electromagnetic field, see Figure 2b. Thus the spatial extension of the EM field of the surface plasmon [31] and strength of a plasmonic mode within the capture layer [32], assuming an exponential decay *Ss* can be expressed as the bulk refractive index sensitivity, given below. This relationship is the basis of LSP resonance wavelength-shift sensing experiments:
(8)$$S_{S}=S_{B}exp\left(\frac{-2z_{0}}{Z_{spp}}\right)\left[1-exp\left(\frac{-2z}{Z_{spp}}\right)\right]$$
where the parameters are defined in Figure 3.

For a plasmonic affinity sensor, (PAS) the need is to aim for the highest response for the target molecule, a goal not achievable using conventional SPs, due to the spatial extent of the field into the surrounding media way above the recognition and target molecules.

Due to these limitations and the active pursuit of higher sensitivity and lowering the limits of detection to measure ultra-low concentrations of substances [32] particularly in bio-applications [33], another classification of surface plasmons found favour, called localised surface plasmons (LSPs). In these devices, the electric field of the SP is in close proximity to the surface of the sensor, to the metal surface, thus increasing the *S_s_*. 

The development of LSPs is very much linked to the improvement of fabrication techniques of nano-patterning of the top metal surface [34]. The excitation of a single LSP is strongly dependent upon the shape and size distribution of nano-patterned metal on a supporting surface topology. To increase the LSP spectral shift, various types of nano-pattern arrays such as nano-spheres, nano-wells and nano-antenna have been fabricated via either chemical or laser lithography [35,36]. High LSP sensitivity therefore demands an efficient nano-design with a simple fabrication technique. A myriad of various nanostructured materials; nano-antennas or nano-wires, dimers or spheroids have been developed and investigated with regards to the plasmonic properties. Physically, they have the intrinsic ability to produce significantly enhanced and highly confined electromagnetic fields and support a highly efficient, LSP resonance [37]. Since those embryonic days, the interest in optical nano-antenna arrays has progressively increased because of the quantum size effect resulting from the discrete electron bands. The modelling of the LSP generated by the various nanostructures is complex. The optical properties of the LSP are influenced by the size, shape and dielectric properties of the nanostructured materials [38,39,40]. For example for nano-spheriods structures, assuming the size of the spheroid is significantly smaller than the wavelength of extinction then the approximate condition *a/*λ *≤* 0.1 applies. The predicted extinction spectrum of the metal composite spheroid as a function of wavelength is given by:
(9)E(λ)=24πNAa3ε(λ)SR32λIn(10)[Im(ε(λ)ms)(Re(ε(λ)ms)+χε(λ)SR)2+Im(ε(λ)ms)2]
where *ε*(*λ*)*_ms_* is the complex dielectric function of the metal composite spheroids, *ε*(*λ*)*_SR_* is the complex dielectric function of the surrounding medium of the sensing platform, *a* is the mean radius of spheroids, *N_A_* is the real density of the nano-blocks/spheroids [38,39]. The other variable is χ, which takes into account the geometry of the spheroid is solved analytically, χ can be defined as [39]:
(10)$$\chi =-1-2{\left[\theta_{0}{}^{2}-\frac{\theta_{0}\left(\theta_{0}{}^{2}+1\right)}{2}\cos^{-1}\left(\frac{\theta_{0}{}^{2}-1}{\theta_{0}{}^{2}+1}\right)\right]}^{-1}$$
the variable *θ*_0_ defined as θ0=(Ymean2Xmean2−1)−12.

Using the above analytical expressions for the extinction of LSP Figure 4 shows the predicted spectrum for various spheroids dimensions and the effect on the extinction spectrum. Also Figure 4 shows the various spheroids shape. 

The authors would like to emphasise that this is only one example, a fuller explanation and methodology to model the LSP can be found in [38,39,40].

## 3. Optical Platforms and Various Strategies Applied to Create SPR Sensing

There are a range of different strategies to create surface plasmons, a number of these are in the planar configuration; lab-on-chip or integrated planar optics and are widely used for biosensors [41]. Here, is a discussion on optical fibre platforms, so planar devices are not explored to any great degree but to make useful comparisons. Furthermore, here for the first time the term "biosensors” is being used, which will be discussed in a little more detail later. 

There are several optical fibre configurations investigated that create surface plasmons that have been used in the past and at present. These can be divided into four major classifications; one being the using the various geometries, such as, fibre tapers, cladding off fibre and end-face mirror/coatings. These set of devices are used to create typically conventional surface plasmons [42], see Figure 5a–c. A second classification is those plasmonic fibre devices that use a grating to couple the light to a metal coating. Various types of optical fibres used in conjunction with gratings, such as, D-shaped fibres or normal cylindrical fibre. These grating plasmonic optical fibre normally produce conventional or damped surface plasmons, see Figure 5d,e. Three is the use of nano-structured coatings, which are effective low-dimensional patterned materials the coupling occurs by several methods [43]. These plasmonic fibre devices create localised surface plasmons [18] or can create a super-structured localised surface plasmons [44], see Figure 5f. Fourthly, is the use of photonic crystal fibre (PCF) [45], there are a myriad of geometric configurations of PCFs. These types of optical fibre product various propagating modes of light that can be close to the surface of the fibre and thus able to excite surface plasmons on the metal coatings, see Figure 5g. Furthermore, the metal coatings that support the surface plasmons can be desposited on various surfaces of the PCF including on the surfaces of the holes inside the PCF fibre, see Figure 5g.

There are metal coated biconical fibre tapers, see Figure 5a, the tapers are classified into two types adiabatic and non-adiabatic. This can be thought of as the degree of coupling that occurs between the tapered core mode and the tapered cladding modes, this is adiabatic and coupling between all supported modes in the taper, such as, tapered cladding to cladding modes. The Slowness Criterion [45] is used to determine the tapers coupling abilities, the magnitude of the gradient of the radius of the fibre along the taper should be significantly smaller than the beat length between the two modes with the closest propagation constants [46]. Otherwise strong mode coupling is to be expected and yielding non-adiabatic mode evolution. An example of an adiabatic biconical tapered optical fibre transmission spectrum with a silver coating of 28 nm, Figure 5a:
(11)$$\left|\frac{\partial r}{\partial z}\right|<<\frac{r}{z_{b}},\;\mathrm{with}\;z_{b}=\frac{2\pi }{\beta_{1}-\beta_{2}}$$

Here *z_b_* is the beat length between two modes with propagation constants *β*_1_ and *β*_2_ and *r* is the radius of the fibre along the taper, *r*/*z_b_* is referred to as the adiabatic length-scale criteria.

The interrogation of tapered fibre devices is relatively simple, consisting of a light source (coherent or non-coherent) and a photodiode detector. Table 1 shows typical performance data for a number of these devices. It needs to be emphasised that the sensitivities quoted are dependent on the refractive index of the materials being studied ranging from 2 × 10^3^ to 50 × 10^3^ nm/refractive index unit (RIU). For example, the sensitivity quoted, 2 to 10 × 10^3^ nm/RIU covers the aqueous index regime. Key advantages of tapered fibre sensors are the simplicity of fabrication and the myriad of configurations available to tailor the spectral properties. Disadvantages are the reproducibility of fabrication, as small changes in the taper profile can significantly change the optical properties and the devices are mechanically fragile leading to issues in packaging. A review of these devices is given in reference [47].

Table 2 shows data for a range of devices that have had the fibre cladding stripped away and the metal coating applied directly on to the fibre core, generally using multimode fibres. The modes spectral characteristics change dramatically with the addition of a metal coating [54]. The close proximity of the metal coating and the fibre core leads to higher coupling efficiency as the E-field of the core modes move spatially closer to the metal. It needs to be emphasised that the core modes change significantly, from guided modes to leaky and radiative modes, [55] thus making modelling more complicated. Furthermore, the movement of the fibre also affects the polarisation condition, which in turn affects the coupling, causing intensity fluctuations leading to measuring instabilities. 

Furthermore, the spectral features are relatively broad, wavelength shifts of a few tens of nanometers, giving rise to a low “figure of merit”. Despite these shortcomings, these sensors have found favour because of their relatively simple manufacturing process and with multiple coating layers, provide relatively high spectral sensitivities [61] but usually spectral index sensitivity of a few 103 nm/RIU, see Table 2. 

Table 3 shows the data for a third group of fibre-plasmon sensors, namely fibre end- face coated mirror devices, see Figure 5c. There are a number of different configurations, some of which use single- mode fibres, others multimode. They can be subdivided into coated flat- end, tip/cone end- section and angle probe. The optical pumping of the multimode flat-end devices can be conceptually treated as free-space systems [65,66,67,68,69]. The fibre modes have a range of propagation constants that allow coupling to surface plasmons in such a way that the modelling procedure given in early part of section two is applicable. Although the sensitivity of these sensors is quite low, their attraction lies in ease of fabrication and the low-cost of interrogation schemes. The sensitivity of these devices can be improved by using additional coatings [70]. Multimode fibres are known to be sensitive to mechanical vibrations or movement and as such, the output signal may fluctuate with time. The surface plasmons generated by the tip/cone version depend heavily on the tip/cone geometry which can be fine-tuned to enhance the spectral sensitivity [68,70] but does make fabrication more difficult. These devices do have mechanical robust issues with handling and operation.

Another family of fibre plasmonic sensors are those that use single-mode fibre in which a grating has been “written” that causes cladding modes to be excited which in turn generate plasmons in the metal coating, see Figure 5d,e. For example, a Bragg grating, a corrugation of the refractive index permanently embedded in the fibre core by means of high-intensity UV radiation. This type of device was first experimentally demonstrated by the authors in 2006 using a tilted fibre grating coated with a uniform 35 nm film of silver [71].

The coupling mechanism is the result of the scattering angles associated with the propagation constants of the various transverse modes (TE/TM, leaky) in the optical fiber and the grating structures coated with a metal. The scattering angle α is calculated from the effective index of the core/cladding modes *n**_β_* by the relationship given by the ray approach; cos(α)=*n**_β_*/*n_cl_*, where *n_cl_* is the refractive index of the cladding, this angle being to the fiber axis. These angles are used to give an associated incidence angle *θ* of each cladding mode onto the metal/ dielectric interface and thus the cladding mode wavenumber projection onto that interface. SPs are generated when this wave-number projection satisfies the dispersion relation of the plasmons given Equations (1) and (2), along with replacing the prism refractive index *n_p_* with the refractive index of the cladding *n_cl_*. Thus firstly, the *n**_β_* and their propagation constants need to be calculated by using a dispersion relationship derived from optical fibre waveguide analysis, such as, Yeh’s algorithm method in conjunction with conformal mapping technique if the geometry of the optical fibre is non-cylindrical. Once the supported modes propagation constants are determined an estimation of the strength of coupling can be obtained by inspecting the overlap integral of the TE/TM fields of the supported modes and the surface plasmons TE/TM fields [72].

The data shown in Table 4, illustrates the performance of three types of grating structures, uniform, tilted, long period, the effect of using different coatings and the refractive index regimes being studied [72,73,74]. The spectral sensitivities range from low 10s nm/RIU to 100s nm/RIU. It needs to be noted that the spectral sensitivities of these plasmonic devices are, in general, higher than those previously covered in this review and are operated in the infra-red rather than the visible, resulting in an overall improvement in performance [14]. The relatively higher spectral sensitivities and narrow spectral width improves the resolution to 10^−8^ RIU^−1^ leading to an increase in FOM. Furthermore, as these devices are mechanically robust the risk of problems in practical applications is minimised. On the downside, fabricating gratings structures is a complex process and grating structures are temperature sensitive, particularly long period gratings. Although this can be compensated for, the interrogation methods are more demanding.

The penultimate classification is the last conventional optical fibres and is the review of the performance of localised surface plasmons (LSP) devices in which the LSPs are generated by means of a formed nanostructured pattern in the thin film coating [18] by means of a periodic strain field set up during fabrication of the sensor, see Figure 4f and [44]. The optical coupling mechanism is essentially the same as discussed previously. Table 5 indicates that the spectral sensitivity can exceed 10^4^nm/RIU and with higher resolution than those of other devices [34]. The spectral features of LSP devices are quite broad, a few tens of nanometers, limiting the resolution attainable to a few times 10^−6^ RIU^−1^. Despite these shortcomings, there has been a surge of interest in using LSPs in chemical and biological detection by means of recognition molecules [81,82,83]. A major issue with these devices is the reproducibility and disorder occurring during fabrication of a few mm [84] and long range disordering of 10 um [85,86]. This long range disordering, these being, size and shape imperfections affects the sensing performance by changing the coupling conditions, excitation wavelength and polarisation dependency [87]. This effect is more so at visible wavelengths. Coupling problems at longer wavelengths (infra-red) results in problems with the interaction length of the plasmons degrading the sensitivity. In addition, polarisation sensitivity is increased [72] leading to measurement instabilities due to mechanical vibrations or movement of the sensor or lead fibre. The other issue with this design of sensing platform is the cost as a result of the complex fabrication process.

Regardless of the above difficulties and issues the LSP yield amongst the best performing ultralow resolution detection schemes for specific molecule species. We will look at this important PAS biosensor applications in a following section of the paper.

The final classification is the use of photonic crystal fibre, there are a myriad of various geometries of fibres being used and investigated and fabricated [45]. At present there seems to be more practical issues to be addressed using PCF as an optical platform for fibre plasmonic sensing devices. These problems include the fabrication, the coating of the films in a controllable manner in the PCF. Furthermore, there are other concerns, such as the delivery of samples to be tested and the connectorisation of the PCF fibre to conventional optical fibres. Considering the issues the PCF optical platform for plasmonic are encouraging yielding some very high spectral sensitivities; 11 × 10^3^ nm/RIU in the index range from 1.33 to 1.41 [97] along with good resolution of 9.1 × 10^−6^ nm/RIU. These may not be amongst the highest but the general overall performance of this class of sensors is good and is encouraging for further investigation, see Table 6.

## 4. Biosensing Applications for Plasmonic Fibre Sensors

Methods In this section we will very briefly look at the importance of and growth in plasmonic fibre biosensors and plasmonic fibre immunosensors. The authors would like to state that is not a review on biosensors but they are acknowledging that this has become dominating area of application for fibre plasmonic sensors, and as such, should be looked at as an individual subject. The first biosensors used unclad fibers and was reported in the literature in 1993 [102]. Interest in these sensors expanded as they had the capability to rapidly analyse and track pathogens, allergens and organic and bio-molecular (i.e., proteins) with high sensitivity and selectivity while allowing to take on-site and in-situ measurements and analysis. These sensors are being investigated for possible use in numerous fields, such as genomics, proteomics, medical diagnosis, environmental monitoring, food analysis and security [43,103]. These sensors have the metal coating onto which is adhered (or the term used is immobilised) a recognition molecule or commonly called a bio-receptor to capture and detect, this is commonly called the functionalisation. A major advantage of such sensors is label-free optical biosensors that can detect in real-time molecular interactions due to the fact that they detect the changes in the refractive index that occurs during the binding interaction of the bioreceptor/receptor with the analyte (the specifics substances in the test sample). There are various types/classifications of receptors used in biosensor, the main types are antibodies, aptamers, self-assembled monolayer (SAM), peptides and affimers, and these are illustrated in Figure 6. All the different receptors have slightly different attributes. 

The most common techniques used are SAM and antibodies. Firstly SAM are used due its simplistic method to functionalise the plasmonic optical fibres by preparing solutions and placing the fibres into capillary tubes, that are sealed to prevent solvent evaporation, followed by washing process and a blocking stage to ensure the biosensor has a high specificity [104,105]. The problem with this method is the limit of detection (LOD), which is higher than other techniques, see Table 7. 

The immobilisation of antibodies is achieved in two ways either electrostatically or by chemical methods, which is more complicated than SAM [106,107] but the LOD performance is better, see Table 6. One of the main issues with DNA antibodies is the shelf life and their “low environmental robustness”, that is to say temperature and liquid-state variations can dramatically affect the performance of the sensor to detect a specific antigen. Furthermore, whilst, in general, the LOD is lower than for the SAM technique there can be issues of chemical selectivity [108,109]. More recently, aptamers are being used more frequently as they are artificially synthesised nucleic acids with benefits such as simple synthesis, high specificity and affinity, good stability, along with a simplistic chemical modification. They have highly selectivity and have been demonstrated detecting substances at ultra-low concentrations; LOD in the attomolar concentrations [110,111]. Furthermore, aptamers are “environmental robust”, have resistance to degradation, have a long shelf life and can be produced in animal-free systems, which have clear ethical and financial benefits. The problem is they are used in conjunction with gold surfaces and the immobilisation can be problematic along with finding the right aptamer for a specific analyte [112]. The remaining two recognition or receptor molecules are peptides, affimers and other associated variations [113,114] the distinct advantage is that they can be immobilised on inorganic surfaces with a significantly greater success than the previous mentioned receptor molecules without using thiol or maleimide chemistry. A common feature of peptides and affimers is that they have an engineered central protein that can be designed for varied recognition/binding surfaces and therefore tuned to recognise different analytes, thus mirroring the behaviour of antibodies [115]. Another advantage of these receptors is that they are smaller than other receptor groups, thus increasing the surface density of the recognition receptor on the biosensor surface increasing the sensitivity. Furthermore, the affimers/peptides layer has a controlled orientation onto the sensor surface and thus yield a greater number of binding surface sites that potentially leads to superior detection properties [116,117]. 

Table 7 shows some typical results for the different recognition/receptor molecules the aptamers technique is showing the ultra-low detection limit [118,122].

## 5. Concluding Remarks

There is one very interesting observation the authors would like to make with regards to the experimental higher sensitivities being achieved and the lowering of detection limits to femto-molar/atto-molar concentrations [122,123] and the theory suggesting the impossibility of approximately subpicomolar detection at the corresponding incubation time. Several authors have looked at this gap between theory and experimental results, suggesting that the theory is looking at the *mean incubation time* whilst the experimental results are based upon the *minimum incubation time* [124,125,126,127]. Furthermore, this approach makes redundant the excessive equilibrium times required to obtain meaningful experimental results. Moreover, as the concentration of the analyte decreases to these ultralow concentration conventional equilibrium behaviour appears to be overtaken by a more stochastic process (*the authors have observed this behaviour experimentally and are working on a paper describing this phenomena*). This leads to several questions on how to interpret the results at these ultralow concentrations and more interesting “at what distance does a molecule feel alone”? Or can we detect a single molecule binding to a receptor in real-time? What we are emphasizing here is that although this field of study has been around for several decades it is entering a new phase of development with great opportunities. 

## Figures and Tables

**Figure 1 sensors-19-04874-f001:**
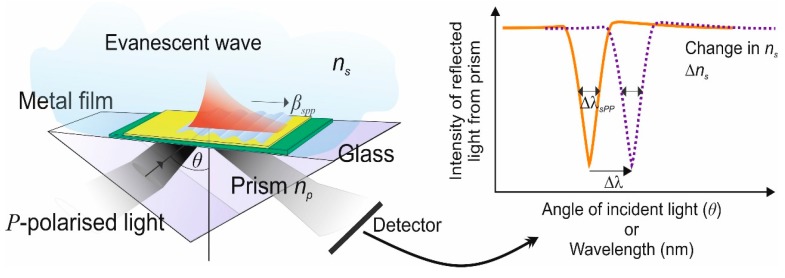
Basic schematic of generation of a surface plasmon resonance.

**Figure 2 sensors-19-04874-f002:**
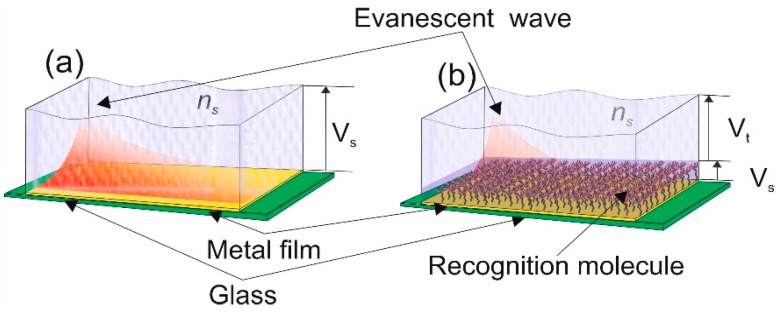
Conceptual illustration of the sensing volume and sensing coverage. (**a**) The full use of the evanescent wave of the surface plasmon to detection of bulk chemical changes in solution. (**b**) the partly use of the evanescent wave of the surface plasmon working in conjunction with recognition molecule adhered to the metal film for specific molecule detection

**Figure 3 sensors-19-04874-f003:**
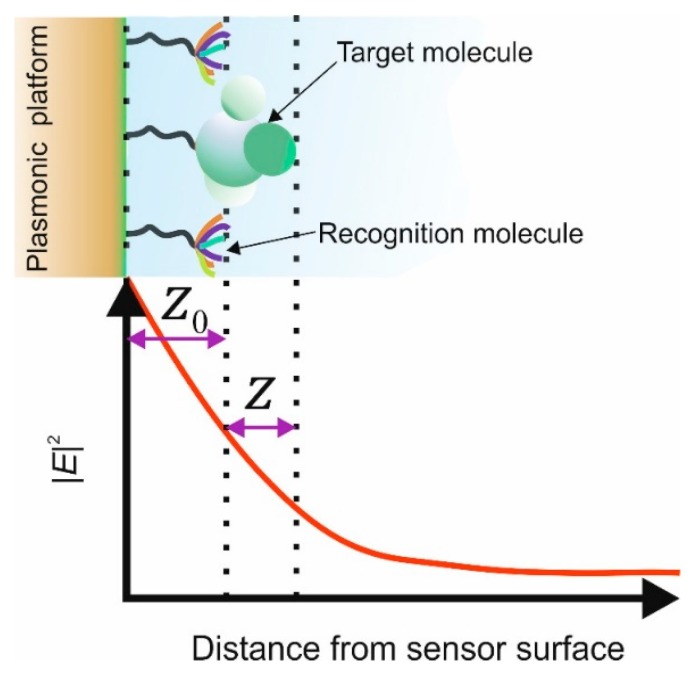
Conceptual representation of the dependence of the surface plasmon electric field on the distance from the metal surface of the sensor.

**Figure 4 sensors-19-04874-f004:**
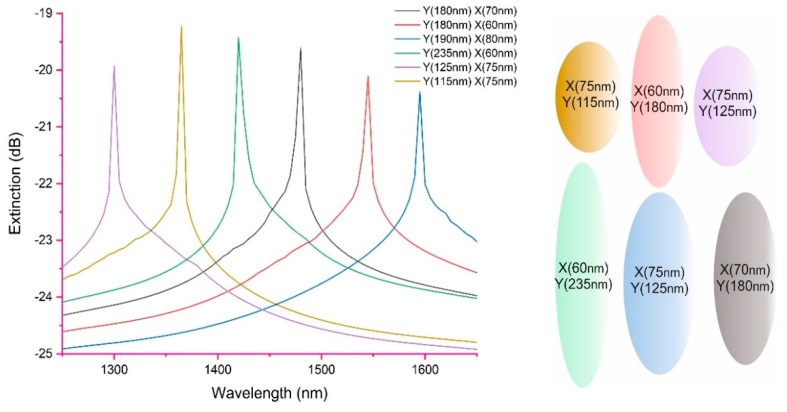
Typical calculated spectral shifts of the extinction of the LSP as a function of the spatial geometry and size of the nano-blocks, using the measured variation in the minor and major axes, for a Ge-SiO2-Au tri-layer with a surrounding index of 1.367. Along with showing the relative changes of shape for the nano-spheriods that generate the spectrums.

**Figure 5 sensors-19-04874-f005:**
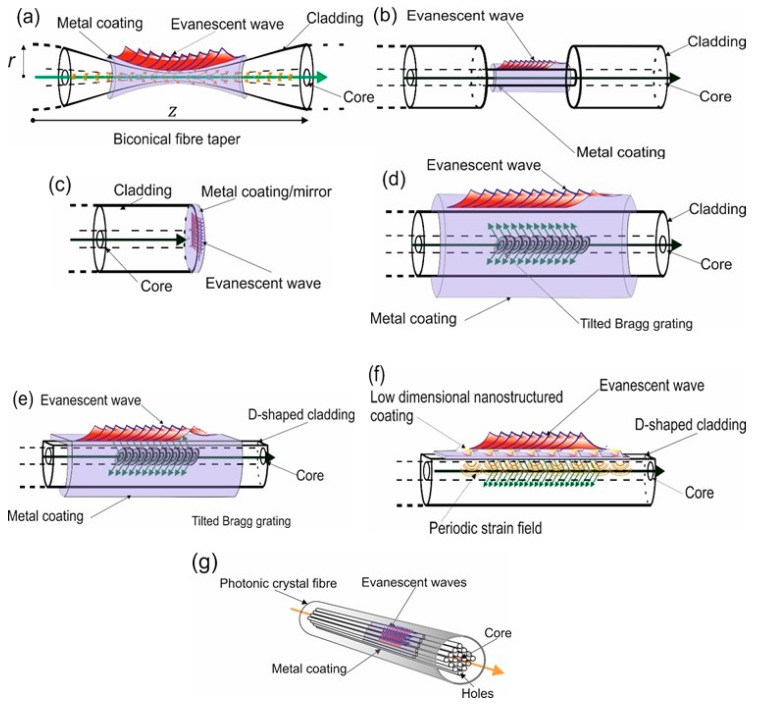
Typical Schematics of various fibre optical platforms used to produce surface plasmons. (**a**) Biconical tapered optical fibre surface plasmon resonance sensor. (**b**) Cladding removed fibre, the coating adhered directly to the core of the fibre. (**c**) An end-face reflection mirror/coating for the generation of the surface plasmons. (**d**,**e**) Conventional cylindrical optical fibre and D-shaped optical fibre, respectively, with a metal coating with the light coupled from a grating structure; tilted. (**f**) A low-dimensional nanostructured material/coating using a periodic strain field as the coupling mechanism for the light. (**g**) Photonic crystal fibre with the metal deposited on the inside of the holes of the fibre which support the surface plasmons.

**Figure 6 sensors-19-04874-f006:**
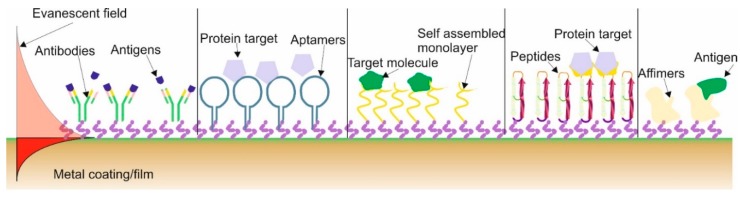
Schematic of a biofunctionalised metal-coated optical fiber surface with the most commonly used strategies to attract specific analytes.

**Table 1 sensors-19-04874-t001:** Summary of various fibre tapered plasmonic sensors.

Tapered Fibre Description	Coating (nm)	Sensitivity nm/RIU	Wavelength (nm)	Resolution RIU^−1^	Index Range	Reference
Biconical tapered Localised surface plasmons	Au, 24 ± 3	51	537	3.2 × 10^−5^	1.33–1.40	[48]
Tapered fibre probe mirrored	Au, 200	400	530–920		1.35–1.42	[49]
Biconical tapered	Ag, 55	2100	500–850	5 × 10^−4^	1.326–1.375	[50]
Biconical tapered asymmetric coating	Au, 26	50 × 10^3^	900–1600	1 × 10^−5^	1.44–1.446	[51]
Multi-biconical tapered	Cr, 1.6 and Au, 50	2 × 10^3^	500–1000	2 × 10^−4^	1.335–1.380	[52]
Multimode Biconical tapered	Au, 50	15 × 10^3^	580–700	1 × 10^−5^	1.333–1.343	[53]

**Table 2 sensors-19-04874-t002:** Summary of various fibre cladding-removed plasmonic sensors.

Unclad Fibre Description	Coating (nm)	Sensitivity nm/RIU	Wavelength (nm)	Resolution RIU^−1^	Index Range	Reference
Tip and Nano-patterened, Multimode	Au, 50 (‘triangular’ size Au, 140 nm)	1610	520–750	3 × 10^−4^	1.333–1.346	[56]
Multimode fibre, Two bare sections	Au, 50 and Au, 50 nm+ silicone gel, few mm	1700	600–1400	1 × 10^−4^	1.333–1.35	[57]
Multimode fibre, Two bare sections	Ag, 30–50 and Ag, 30–50	2000	500–850	1 × 10^−3^	1.33–1.38	[58]
Single section multicoatings	ITO, 80 Ag, 40	3 × 10^3^	580		1.33–1.36	[59]
Multimode fibre, Two bare sections	Ag, 40 and Cu, 40		500–840	2.4 × 10^−5^	1.3365–1.5859	[60]
Single section and molecular imprinted polymer	Ag, 40	8 × 10^3^	400–800	3.5 × 10^−5^	1.38–1.385	[61]
Polarisation maintaining fibre exposed core	Au, 55 and Ta_2_O_5,_ 15	3150	770–880	4 × 10^−4^	1.33–1.35	[62]
coreless fibre connected to multimode fibre	Au, 40	2.7 × 10^3^	350–1100	8 × 10^−5^	1.3335–1.4019	[63]
PCF D-shaped with rectangular lattice	Au, 30	8.1 × 10^3^	600–900	1.2 × 10^−5^	1.33–1.41	[64]

**Table 3 sensors-19-04874-t003:** Summary of various fibre end-face plasmonic sensors.

Tip/Probe Fibre Description	Coating (nm)	Sensitivity nm/RIU	Wavelength (nm)	Resolution RIU^−1^	Index Range	Reference
Single mode End-face tip	Au, 40 Ag, 40	51	500–800		Air	[65]
Multimode fibre, Flat core end-section	Au, 65	1557	400–800	3.2 × 10^−4^	1.333–1.349	[66]
Multimode fibre, Flat core end-section	Ag, 55	1722	600–700	5 × 10^−4^	1.4–1.7	[67]
tip end-section	Au, 40 Ag, 40	4 × 10^3^	650	2.5 × 10^−4^	1.33–1.375	[68]
Multimode fibre, Flat core end-section	Ag, 50 and SiO, 2000	1020	400–1000	2.4 × 10^−4^	1.336–1.585	[69]
Single mode End-face tip	TiO_2_, 0–110 and Au 50	2.7 × 10^3^	590–720	8 × 10^−5^	1.333–1.383	[70]

**Table 4 sensors-19-04874-t004:** Summary of various fibre grating plasmonic sensors.

Grating Fibre Description	Coating (nm)	Sensitivity dB or nm/RIU	Wavelength (nm)	Resolution RIU^−1^	Index Range	Reference
Single mode fibre small angled FBG, D-shaped	Au, 32	3365	1220–1700	5 × 10^−6^	1.30–1.38	[72]
Single mode fibre with LPG	Au colloids, 8.4 ± 2.8	23	1510–1550	3.2 × 10^−4^	1.33–1.4344	[75]
Single mode fibre, angled FBG	Cr, 2–3, Ag, 20–50	8100	1440–1550	1 × 10^−5^	1.34–1.3408	[76]
Single mode fibre with LPG	Au, 20–50	12	653–667	1 × 10^−5^	1.33–1.38	[77]
Single mode fibre small angled FBG,	Au, 50	1000	1450–1560	1 × 10^−4^	1.3363–1.3364	[78]
Single mode fibre small angled FBG,	Au, 10–30	470	1520–1560	6 × 10^−5^	1.3335–1.4019	[79]
Single mode etched fibre small angled FBG,	Cr, 2 and Au, 30	510	1520–1590	6.8 × 10^−5^	1.335–1.432	[80]
Single mode fibre mid-range angled FBG,	Au, 50	5515	1280–1560	1 × 10^−8^	1.30–1.44 and Air	[34]

**Table 5 sensors-19-04874-t005:** Summary of various nanostructured and localised plasmonic sensors.

Material Description	Coating (nm)	Sensitivity dB or nm/RIU	Wavelength (nm)	Resolution RIU-1	Index Range	Reference
Single mode fibre Multilayers, surface metallic grating	Ag, 43.3 and Au 1.2 nm	4000	400–1000	5 × 10^−5^	1.33	[88]
Single mode fibre D-shaped Multilayered ordered nanowires	Ge, 48 and SiO2, 48 and Pt, 36	1 × 10^4^	1260–1680	3.2 × 10^−4^	1.33–1.39	[89]
Multimode fibre, End-face nanospheres	Au, 35	2700	400–700	3.7 × 10^−6^	1.3–1.38	[90]
Single mode fibre Tilted FBG random nanowires	Ag, 20–50	175	1520–1610	1.6 × 10^−5^	1.33–1.365	[91]
Single mode fibre Tilted FBG random nanowires	Au, 50	651	1520–1600	1.5 × 10^−5^	1.330–1.347	[92]
Single mode fibre D-shaped Multilayered ordered nanowires	Ge, 48 and SiO_2_, 48 and Ag, 32	1.2 × 10^4^	1250–1680	1 × 10^−6^	1.3–1.39	[44]
Single mode fibre Tilted FBG nanowires	Ag, 30	1000	1520–1600	1 × 10^−5^	1.330–1.345	[93]
Single mode D-shaped etched to core fibre nanostrips	Au, 20	2 × 10^4^	1400–1680	2.3 × 10^−6^	1.327–1.333	[94]
Single mode fibre bioconical taper, nanospheres	Au, 10.5	6000	400–1000	1 × 10^−5^	1.35	[95]
Multimode U-fibre, Triangular nanoparticles	Ag, 80–100	1116.8	300–600	9 × 10^−6^	1.334–1365	[96]

**Table 6 sensors-19-04874-t006:** Summary of photonic crystal fibre plasmonic sensors.

PCF Fibre Description	Coating (nm)	Sensitivity nm/RIU	Wavelength (nm)	Resolution RIU^−1^	Index Range	Reference
Irregular cross-section with core and adjacent hole	Au, 30 to 50	11 × 10^3^	770–1070	9.1 × 10^−6^	1.33–1.41	[97]
Multimode Micro fibre, Two bare sections and circular PCF	Au, 50 and Graphene	4649	400–1000	1 × 10^−5^	1.33–1.37	[98]
Horse-shoe shaped PCF	Ag, 30–40 and graphene	2290	500–700	5 × 10^−5^	1.33–1.3688	[99]
D-shaped Endless Single Mode PCF	Au 45	7381	560–660	5 × 10^−5^	1.40–1.42	[100]
D-shaped Endless Single Mode PCF	Au 45	2336	560–660	6.5 × 10^−5^	1.33–1.37	[101]

**Table 7 sensors-19-04874-t007:** Examples of various optical plasmonic biosensors.

Optical Platform	Functional Material	Sensitivity dB or nm/RIU	Analyte	Limit of Detection	Reference
Single mode fibre long period grating LSPR	Self-assembled Au colloids + dinitrophenyl compound (DNP)	23.45	anti-DNP	950 pM or 1.4 × 10^−7^ g/mL	[117]
Multimode optical fiber probe bundle, endface LSPR	Au-capped nanoparticle layer+ peptide nucleic acids		PNA-DNA	0.677 pM	[118]
Single mode taperd fibre, End-face nanospheres	Au coating+Aptamer (short chain oligonucleotide)		17β-Estradiol	2.1 nM or 0.6 ng mL^−1^	[119]
Multimode fibre, nanopatterned coating LSPR, End-face	Au coated antibodies		Prostate-specific antigen	0.1 pg mL^−1^	[120]
Superchiral plasmonic nanostructure, planar	Affimer amino acid (G65C and K67C)	400	G65C and K67C		[121]
Single mode lapped fibre with nano-structured thin film LSPR	Au + thrombin aptamer 5′ – SH(CH2)6 - GGT TGG TGT GGT TGG - 3′	3.4 × 10^4^	A thrombin	50 aM	[122]

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
