# Peer review of "A Review: Evolution and Diversity of Optical Fibre Plasmonic Sensors"

_sensors, 2019, doi:10.3390/s19224874_

Round 1
Reviewer 1 Report
The paper presents the review of plasmonic fiber-optic sensors. The manuscript is well written and can be published after minor corrections.
1. Page 3, line 120: missing Greek symbol for permitivity (epsilon).
2. Starting from page 3: no equation labels.
3. Page 4, line 142: missing Greek symbol for wavelength (lambda).
4. Variables names should be typed with italic font:
Page 4, Line 174;
Page 5, Lines 191—193, 201, 219.
5. Page 8, Line 325: Repeated text "ranging from".
6. Page 8, Line 325: There is no definition of RIU in text.
Author Response
To the Reviewers and Editor of the manuscript entitled “A Review: Evolution and Diversity of Optical fibre Plasmonic Sensors”
Firstly, both authors would like to thank all three reviewers for their efforts and comments to improve the quality of the review paper.
General comment:-
We have taken the points made and have altered the review paper accordingly:-
(1) Changing the abstract
(2) Adding new text to the introduction
(3) Adding new text in section 3
(4) Added a new figure; figure 5g
(5) Added new section on PCF plasmonic sensors
(6) Added a new Table; Table 6.
(7) Added 6 new references.
Specifically to the comments of the Reviewers:-
Reviewer 1 comments
The paper presents the review of plasmonic fiber-optic sensors. The manuscript is well written and can be published after minor corrections. Ans. Both Ron and I would to thank the Reviewer for is comments and “well-spotted” the mistakes. Page 3, line 120: missing Greek symbol for permitivity (epsilon).
Ans. DONE
Starting from page 3: no equation labels.
Ans. DONE
Page 4, line 142: missing Greek symbol for wavelength (lambda).
Ans. DONE
Variables names should be typed with italic font:
Page 4, Line 174;
Page 5, Lines 191—193, 201, 219.
Ans. Done
Page 8, Line 325: Repeated text "ranging from".
Ans. DONE
Page 8, Line 325: There is no definition of RIU in text.
Ans. DONE
Yours Sincerely
Thomas Allsop and Ron Neal.
Reviewer 2 Report
Congratulations on a clear, informative and through review of fiber optic-based SPR sensors. This is an excellent overview and, no doubt, a reference tool for many involved with FO sensors and associated devices.
Author Response
To the Reviewers and Editor of the manuscript entitled “A Review: Evolution and Diversity of Optical fibre Plasmonic Sensors”
Firstly, both authors would like to thank all three reviewers for their efforts and comments to improve the quality of the review paper.
General comment:-
We have taken the points made and have altered the review paper accordingly:-
(1) Changing the abstract
(2) Adding new text to the introduction
(3) Adding new text in section 3
(4) Added a new figure; figure 5g
(5) Added new section on PCF plasmonic sensors
(6) Added a new Table; Table 6.
(7) Added 6 new references.
Specifically to the comments of the Reviewers:-
Reviewers 2 comments
Congratulations on a clear, informative and through review of fiber optic-based SPR sensors. This is an excellent overview and, no doubt, a reference tool for many involved with FO sensors and associated devices.
Ans.
Both Ron and I would like to thank the kind words of reviewer 2
Yours Sincerely
Thomas Allsop and Ron Neal.
Reviewer 3 Report
The manuscript presents a review of the evaluation and diversity of plasmonic fibre optic sensors. Various strategies of SPR sensing and their limitation are described after explaining the underlying physics of SPR. A good set of plasmonic sensor examples are listed which definitely provides a collective overview of this research area. However, the manuscript fails to justify the necessity of such a review paper. Example of similar reviews are but not limited to
(1) M.S.A. Gandhi et al., "Recent Advances in Plasmonic Sensor-Based Fiber
Optic Probes for Biological Applications", Applied Sciences (MDPI), 2019
(2) A.K. Sharma et al., "Design and Performance Perspectives on Fiber Optic Sensors With Plasmonic Nanostructures and Gratings: A Review", IEEE Sensors Journal, 19, 17, 2019
The title of the manuscript creates the notion that it is going to outline all kinds of possible fibre optic sensors. However, the Abstract section only mentions the FBG and SPR based fibre sensors. The mention of the funding source in the abstract to my humble opinion is a failed attempt of attacting the attention of the readers. Description of relatively new fibre sensors (e.g. photonic crystal fibre) is absent in the manuscript. Some important aspects (e.g. bending loss, amplitude sensitivity etc.) and their dependence on material and structural properties are also not discussed. Overall, the work fails to create new knowledge from the review of research endeavours in this area to date.
Other minor comments:
Pg 7, ln 306: spelling mistake. Correct: "..light coupled to form a grating.."
pg 8, ln 325: 'ranging from' used twice
Some recommendation which I believe would make the paper more valuable:
(1) Design methodology of different fibre sensors, not to mention the already described FBG and SPR based ones
(2) A comparison of different sensors in terms of ease of fabrication and implementation costs
(3) Inclusion of other applications with descriptions, as was section 4. Biosensing applications...sensors.
Author Response
To the Reviewers and Editor of the manuscript entitled “A Review: Evolution and Diversity of Optical fibre Plasmonic Sensors”
Firstly, both authors would like to thank all three reviewers for their efforts and comments to improve the quality of the review paper.
General comment:-
We have taken the points made and have altered the review paper accordingly:-
(1) Changing the abstract
(2) Adding new text to the introduction
(3) Adding new text in section 3
(4) Added a new figure; figure 5g
(5) Added new section on PCF plasmonic sensors
(6) Added a new Table; Table 6.
(7) Added 6 new references.
Specifically to the comments of the Reviewers:-
Reviewers 3 comments
The manuscript presents a review of the evaluation and diversity of plasmonic fibre optic sensors. Various strategies of SPR sensing and their limitation are described after explaining the underlying physics of SPR. A good set of plasmonic sensor examples are listed which definitely provides a collective overview of this research area. However, the manuscript fails to justify the necessity of such a review paper. Example of similar reviews are but not limited to
Point 1, I would like to make:-
The title of the paper is actually “ A Review: Evolution and Diversity Plasmonic Fibre Optic Sensors”
But have changed this slightly to “A Review: Evolution and Diversity of Optical fibre Plasmonic Sensors”
It’s the evolution and not evaluation..
(1) M.S.A. Gandhi et al., "Recent Advances in Plasmonic Sensor-Based Fiber
Optic Probes for Biological Applications", Applied Sciences (MDPI), 2019
(2) A.K. Sharma et al., "Design and Performance Perspectives on Fiber Optic Sensors With Plasmonic Nanostructures and Gratings: A Review", IEEE Sensors Journal, 19, 17, 2019
Ans. The reviewer has made a good point and we address this in the abstract and the final paragraph of the introduction.
The title of the manuscript creates the notion that it is going to outline all kinds of possible fibre optic sensors. However, the Abstract section only mentions the FBG and SPR based fibre sensors. The mention of the funding source in the abstract to my humble opinion is a failed attempt of attacting the attention of the readers. Description of relatively new fibre sensors (e.g. photonic crystal fibre) is absent in the manuscript. Some important aspects (e.g. bending loss, amplitude sensitivity etc.) and their dependence on material and structural properties are also not discussed. Overall, the work fails to create new knowledge from the review of research endeavours in this area to date.
Ans.
The title is looking at fibre plasmonic fibre sensors, not all fibre sensors. The abstract has been changed, the mention of the EPSRC is not an attempt to grab reader’s attention, we are just being factual. So we are not misunderstood the abstract has been rewritten. We would like to thank the reviewer and now we have included PCF plasmonic fibre sensors, new Table 6 and six new references. We did state that we would only consider one sensitivity for a comparison but we do mention other optical parameters are used for sensing. “… their dependence on material and structural properties…”This is not true this is intrinsically included in the modelling of the waveguide used as the optical platform for the surface plasmons and we do use the optical material constants in the modelling procedure 1 to 3, 6, 9 and 10.
Other minor comments:
Pg 7, ln 306: spelling mistake. Correct: "..light coupled to form a grating.."
Ans. DONE
pg 8, ln 325: 'ranging from' used twice
Ans. DONE
Some recommendation which I believe would make the paper more valuable:
(1) Design methodology of different fibre sensors, not to mention the already described FBG and SPR based ones
Ans. General methodology is given for all the sensors. Furthermore, the methodology this is very much interwoven with the applications and references are given. The two authors are giving an overall view and approach help readers.
(2) A comparison of different sensors in terms of ease of fabrication and implementation costs
Ans. The reference are given and the reader to make their own minds up with regards to “ease of fabrication”. Concerning “costs” this is very much dependent upon the resources available to the individual researchers.
(3) Inclusion of other applications with descriptions, as was section 4. Biosensing applications...sensors.
Ans. The general spectral sensitivity and spectral characteristics are the underline parameters for sensing thus looking at these parameters are very useful for researchers for all applications. Biosensing is looked at specifically due to the high interest of using fibre optic plasmonic sensor as bisosensors are the present time.
Yours Sincerely
Thomas Allsop and Ron Neal.
Round 2
Reviewer 3 Report
I would like to thank the authors for addressing the reviewer comments.